# Evaluation of Forest Edge Structure and Stability in Peri-Urban Forests

**David Hladnik** [1] , **Andrej Kobler** [2] **and Janez Pirnat** [1,*]

1   Department of Forestry and Renewable Forest Resources, Biotechnical Faculty, Department of Forestry and Renewable Forest Resources, University of Ljubljana 1000 Ljubljana, Slovenia; david.hladnik@bf.uni-lj.si
2   Slovenian Forestry Institute, 1000 Ljubljana, Slovenia; andrej.kobler@gozdis.si
*   Correspondence: janez.pirnat@bf.uni-lj.si

**Abstract:** In the presented research, we studied the forest edge structure of urban and peri-urban forests on the outskirts of Ljubljana (Slovenia) consisting of a number of patches covering the collective surface of 1884 ha. They differ from each other according to the degree of fragmentation and by the share of the interior forest area. On the basis of LiDAR data, we conducted an analysis of the edges of the persistent forest patches and estimated them with regard to the land use they bordered on. The horizontal estimation of forest edges and the changes of forest edges, in the last decades, were estimated using digital orthophoto images of cyclic aerial surveys of Slovenia, from 1975 to 2018. The data, provided by LiDAR, were used to obtain an accurate estimate of forest edges and the metrics of their vertical canopy structure. On the basis of the canopy height model (CHM), we determined the height classes, the heights of the tallest trees, and indices of canopy height diversity (CHD) as variables subjected to a *k*-means cluster analysis. To determine the forest edge and trees stability, their heights and diameters at breast height (DBH) were measured and their canopy length and *h/d* (height/diameter) dimension ratios were estimated. In the study area of the Golovec forest patch, more than half of the forest edge segments (56%) border on residential buildings. After the construction of buildings, 54% of the newly formed forest edges developed a high and steep structure. Unfavorable *h/d* dimension ratio was estimated for 16% of trees, more among the coniferous than among the deciduous trees. Similar characteristics of newly formed forest edges bordering on built-up areas were determined in other sub-urban forest patches, despite the smaller share of such forest edges (19% and 10%, respectively). Tools and methods presented in the research enable the implementation of concrete silvicultural practices in a realistic time period and extend to ensure that adequate forestry measures are taken to minimize possible disturbances.

**Keywords:** forest ecosystem services; forest edge stability; forest edge structure; LiDAR; urban forests

## 1. Introduction

In the past, European landscapes changed most significantly under the influence of agriculture, whereas since the end of the 20th century, they have been increasingly affected by urbanization [1–3]. Urban and sub-urban forests are exposed to the pressure of urban expansion, whereas, at the same time, they represent an important source of health and well-being of urban inhabitants owing to their many beneficial effects [4,5]. Urban and sub-urban forests fulfil many different functions, but contradictions can also sometimes arise, especially between the so-called provisioning services and regulating services [6]. The forest functions themselves can also change, both in the extent and in the intensity of use, for which two reasons have been established. The first is the changing human need for ecosystem services [7], while the second is new findings and tools that enable a more accurate spatial and contextual definition of ecosystem services [8].



So far, researchers have mainly focused on the ecosystem services of urban forests. The directions of sustainable management preserving ecosystem services are crucial for the process of forest management. Less attention has been paid to the management of forest edges in urban areas which represent a visual contact with nature for the urban population [9,10] and form a transition zone between forest and open land in various patterns and distributions between different land uses [11]. Science-based guidelines have been developed to facilitate the planning and designing of conservation buffers in rural and urban landscapes [12]. Some authors, additionally, call attention to possible negative impacts of trees and forests (ecosystem disservices) in urban areas [13–15]. Measuring to what extent urban green spaces represent harmful, unpleasant, or unwanted annoyances, classified as disservices to urban dwellers, provides a good starting point for the analysis of green spaces and peri-urban forests in contact with urban areas. Among the most frequently mentioned disservices are obstruction of traffic on roads and pavements, displeasure with messiness and clutter, and the risk of trees falling and causing damage to buildings and property [14]. In urban areas and in the vicinity of urban infrastructure (roads, railways, and power lines), graded forest edges are recommended, as they not only ensure ecosystem services but also limit potential hazards from tall trees at forest edges and alongside the above-mentioned urban infrastructure [11,16].

However, it should be noted that graded forest edges are rare and do not receive as much attention as the forest structures typical of interior forest areas in the forest management process. According to the reports of the national forest inventories at the beginning of the decade, only 12 of 26 European countries collected information on the characteristics of forest edges, in particular their horizontal shape and length [17]. On the basis of the data provided by the Swedish National Forest Inventory (NILS), it has been estimated [18] that only 20% of forest edges have a developed shrub belt and only 2% of forest edges have a graded structure. In the Swiss National Forest Inventory, shrub belt and a stepped, loose shelter belt were not estimated, while the shrub belt in front of the shelter belt was estimated to cover only 5.9% of the forest edges [19]. As in Canada [13], the damage caused by the ice storm to trees in forests and urban green areas has also increased people's awareness in Slovenia of the importance of forest management and cultivation on forest edges [20].

As estimated in previous studies, forest patches in the Ljubljana area have been classified as persistent forests over the last 200 years [21]; even in the last 40 years of intensive urbanization, the edges of these patches have not been significantly altered. The majority of the clearings, which affected only 8.4% of the entire peri-urban forest surface, occurred due to the construction of the highway, while clearings for the benefit of urbanization were carried out primarily as extensions of existing settlements at forest edges [22]. In past centuries, agricultural land was more suitable and accessible than forest land to build on, and natural areas were protected from urbanization. As in the study by Wiström [11] on forest edges in urban areas, these forest edges can be classified as stationary, since edge displacement has not been observed, and therefore is permanent for the time being.

In the presented article, we continue the study of the ecosystem services of urban and peri-urban forests in the city of Ljubljana, Slovenia. In the past, we had already conducted a thorough analysis of the aesthetic, recreation, and diversity functions of the aforementioned forests [21–23]. For the assessment of these functions, we relied mainly on the fieldwork methods and the data on the past land use, spatial distribution, and estimated stand structure.

In this paper, we have focused on the structure of the forest edge which can significantly affect the sustainability of the protective and climatic functions of the forest.

We analyzed the condition of forest edges in the areas of the selected sub-urban forest patches and compared how different land uses and human interventions that lead to forest edge clearings influenced their structure. We compared the differences between the areas of stationary forest edges and clearings in the heart of urban areas, at the edge of a landscape park, and on the outskirts of urban areas, where the persistency of forest edges has been influenced not only by agricultural land use, but also by urbanization and building of infrastructure. Structural characteristics of the forest edges were

assessed using data from airborne laser scanning (ALS), cyclic aerial surveys over a 40-year period, and fieldwork on a systematic sampling grid.

Our aim is to provide a starting point for monitoring systems in sub-urban forest management, which enables observation of the development of forests bordering on urban and agricultural lands and identifies forest edge structure which could lead to unwanted conflicts between forest and urban land use at forest edges.

## 2. Materials and Methods

### 2.1. Study Area

The research study area consists of a 30 × 20 km section encompassing Ljubljana and its vicinity, as well as all important urban and suburban forests discussed in the previous studies [21–23]. The city with 293,000 inhabitants is situated in the flat area of the Ljubljana Basin, 290 m a.s.l. (above sea level), and is surrounded by a hilly forested landscape reaching up to 700 m a.s.l. Forests grow on hilly surroundings, whereas forest patches on the plains are remnants of former hornbeam and oak forests. Among the sub-urban forests of Ljubljana, the following site types are the most prevalent [24]: submontane *Fagus sylvatica* forests on silicate bedrock, acidophilous *Pinus sylvestris* forests, forests of *Quercus robur* and *Carpinus betulus*, forests of *Carpinus betulus* with *Quercus petraea* on silicate bedrock, and forests of *Alnus glutinosa*.

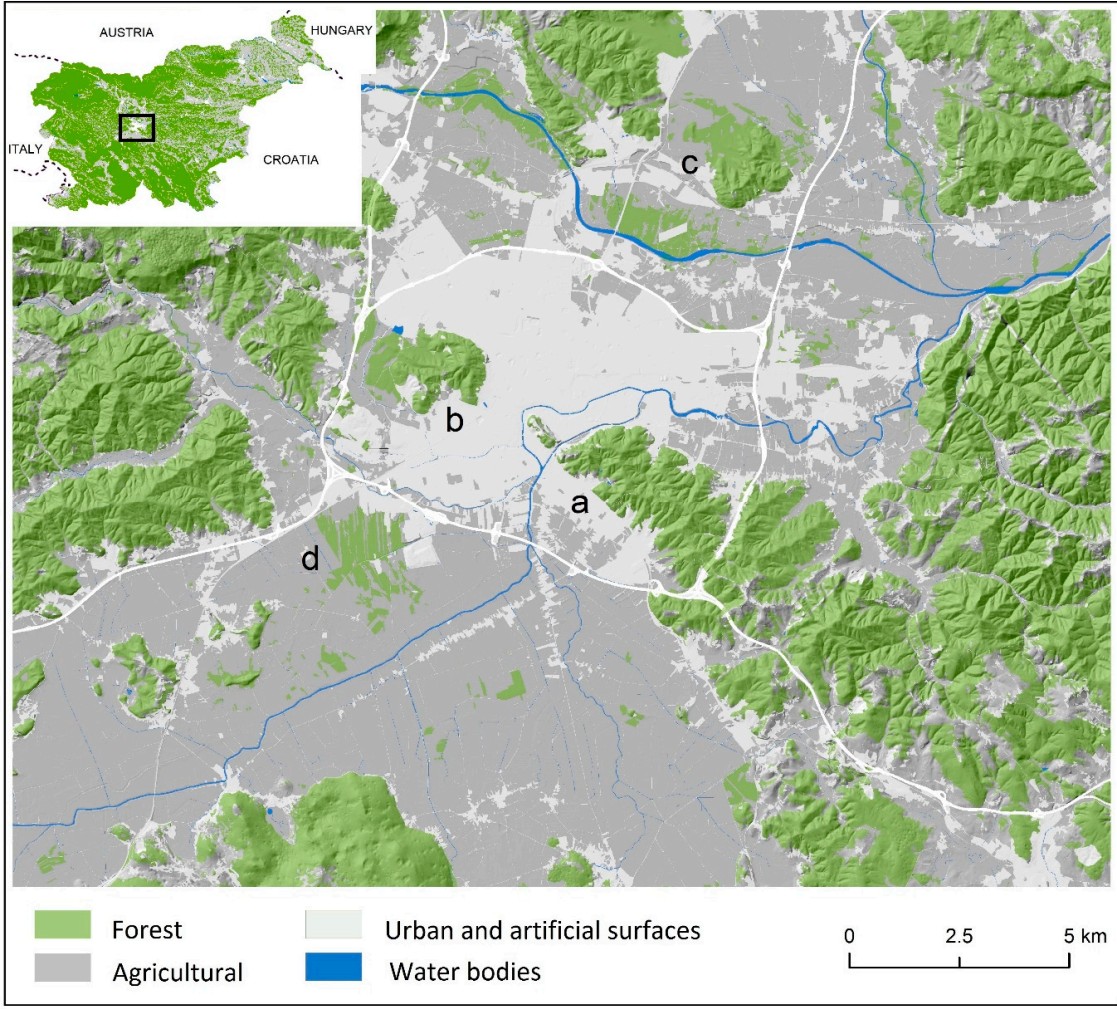

**Figure 1.** The wide study area of Ljubljana with the marked positions of four forest patches on the outskirts of the city. (**a**) Golovec (east); (**b**) Rožnik (west); (**c**) Nadgorica (north); and (**d**) Barje (south).

According to Slovenia Forest Service data [25], the tree species with the highest share of growing stock in the forests of the Golovec forest patch are beech *Fagus sylvatica* (25%), sessile oak *Quercus petraea* (23%), Scots pine *Pinus sylvestris* (22%), chestnut *Castanea sativa* (13%), and spruce *Picea abies* (11%). The Rožnik forest patch represents the majority of a landscape park dominated by spruce *Picea abies* (37%), Scots pine *Pinus sylvestris* (16%), sessile oak *Quercus petraea* (14%), and chestnut *Castanea sativa* (11%). Botch patches Golovec and Rožnik indicate certain green corridor possibilities in the NW–SE direction, thus, representing a link between neighbouring landscape units [21], recognized as a green wedge, formed by Tivoli Park [26].

Nadgorica, in the northern part of the study area, includes the forest comprised of several patches, which are interesting both due to their diversity, as well as their position, as they lie in the immediate neighbourhood of Ljubljana and have emphasized diversity, hydroclimatic, and recreation functions [21–23,27]. The patches are dominated by Scots pine (32%), spruce (21%), black alder (13%), beech (10%) and sessile oak (10%).

In the southern outskirts of the city, most of the lands covered by trees and woods in the neighbourhood of forest patches are classified as non-forest lands consisting of alder coppices and poplar plantations growing on abandoned agricultural lands on the marshes. Due to the regional waste management center situated in the middle, most of the area is unsuitable for visitors. In the western part classified as a forest land, black alder also prevails (46%), followed by pedunculated oak (30%) and spruce (22%). The four described forest patches are also composed of tree species forming the peri-urban forests in the wide study area (Figure 1), as they are typical of the discussed prevalent forest site types [24].

## 2.2. Airborne LiDAR Data Pre-Processing and Delineating of Forest Edges

We estimated the structure of stands and forest edge using LiDAR-recorded data freely available and covering the entire Slovenia [28]. The data are gathered on 1 km$^2$ sheets in the ZLAS format that can be processed in different GIS environments, in our case in the GIS ArcMap 10.4 environment [29]. The data were recorded in March and June 2014 with the Eurocopter EC 120B helicopter flying from 1200 to 1400 m above ground. The LiDAR system was equipped with the RIEGL LMS-Q780 laser scanner with pulse frequency of 400 kHz and with a positioning and orientation system (differential GNSS Novatel OEMV-3, inertial navigation system INS IGI Aerocontrol Mark II. E 256 Hz). The estimated density was 5 points/m$^2$ at the laser beam diameter or a footprint with the size of 30 cm, positional accuracy of 30 cm, and ellipsoidal height accuracy of 15 cm.

In accordance with the technical report on the laser scanning of Slovenia and products based on the scanning data [28], an automatic processing of point cloud data and manual error removal was applied in order to create a digital terrain model (DTM). Later, we adopted the provided data to also form a digital surface model (DSM). We obtained the digital surface model and a digital canopy height model (CHM) with a 1 m spatial resolution by subtracting the DTM from the DSM in the georeferenced and classified LiDAR point cloud. We used the data provided by LiDAR in order to obtain an estimate of forest edges and the metrics of their vertical vegetation structure. For each one-meter cell, we collected data on vegetation height from the canopy height model and determined height classes and vegetation components by height classes above ground level defined as ≥0.5 m. In national forest inventories [30], woody plants higher than 0.5 m are considered to be a part of the shrub belt and dwarf shrubs and woody plants <0.5 m are part of the herb border, similar to the estimation of vegetation structure in fragmented woodlands and their edges [31].

On the basis of the CHM, we determined the following height classes: 0.5 to 4.9, 5 to 14.9, 15 to 24.9, and ≥25 m. These height classes were adopted from FAO [32], which defines forests by the presence of trees and the absence of other predominant land uses. We determined the upper height classes based on the recommended height of the development stages of pole stands, mature trees, and old trees in Central European conditions [33]. We tested the horizontal estimation of forest edges and possible inaccuracies in the delimitation of agricultural land in areas where agricultural vegetation

could exceed 0.5 m in height, based on digital orthophoto images taken in 2018, and spatial data on the evidence of the actual land use of agricultural and forestry land determined from the same orthophoto images [34]. The older panchromatic black-and-white (PAN) aerial film photographs taken at a scale of 1:17,500 were scanned using 21 μm resolution to achieve comparable resolution or ground sampling distance (GSD) of 50 cm. The absolute orientation of the images was based on at least 10 to 15 ground control points. Since we acquired control points from topographic maps, we could only achieve a positional accuracy of 2 to 3 m for orthophoto images. To estimate the horizontal changes of forest edges in the last decades we prepared digital orthophoto images based on the cyclic aerial survey of Slovenia from 1975. In order to determine forest edge changes after 1975, such as clearings and overgrowing, we visually recorded and delimited them with the help of digital orthophoto images from the aerial survey of Slovenia. We used infrared and panchromatic digital orthophoto images to detect changes and to verify the existing forest edges based on the LiDAR data (Figure 2).

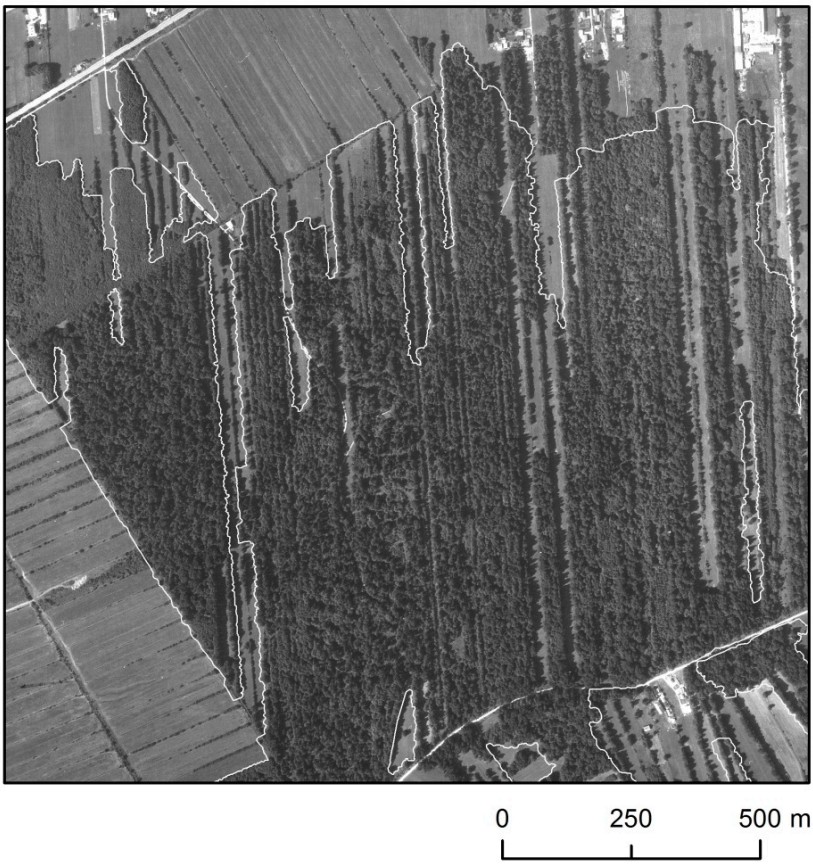

**Figure 2.** Orthophoto image showing forest patches, in the Barje area, in the year 1975, and the present forest edges (white line) determined based on the LiDAR data from the year 2014.

### 2.3. Calculating Variables and Cluster Analysis

The first step in the analysis of forest edges was dividing the lines defining polygons of forest patches into equal, 30 m long segments in the GIS ArcMap environment. Then, we used the segments to determine buffer zones 10, 20, and 30 m from the forest edge towards the interior of forest patches. We also adopted the size of 300 pixels as the smallest segregation unit in the forest edge area. In this way, we acquired groups of stand heights in each of the depth belts of the forest edge. For each patch, we assumed a 30 m depth of the forest edge, as this value approximately corresponds to the forest stand height. According to the visual assessment on the digital orthophoto image, we determined for each of the polygons whether it borders on an agricultural land, road, or power line network, or on a built-up area, if its distance from the edge did not exceed the length of a mature tree (30 m).

We also calculated the depth of the interior area 100 and 300 m from the forest edge for each of the forest patches. For the same distances, we also determined 30 m long and 10 m wide segments. Our aim was to determine whether the canopy structure of forest edges differs from the canopy structure of forest stands in the interior area of forest patches (Figure 3). We selected the distances of 100 and 300 m based on the suggested recommendations argued in the previous studies [22,35]. In several scientific papers, a minimum patch size of 30 to 40 ha has been chosen in studies of habitat fragmentation for the protection of forest-interior species at the landscape scale. A circular patch with such an area would have a radius of at least 300 m [22].

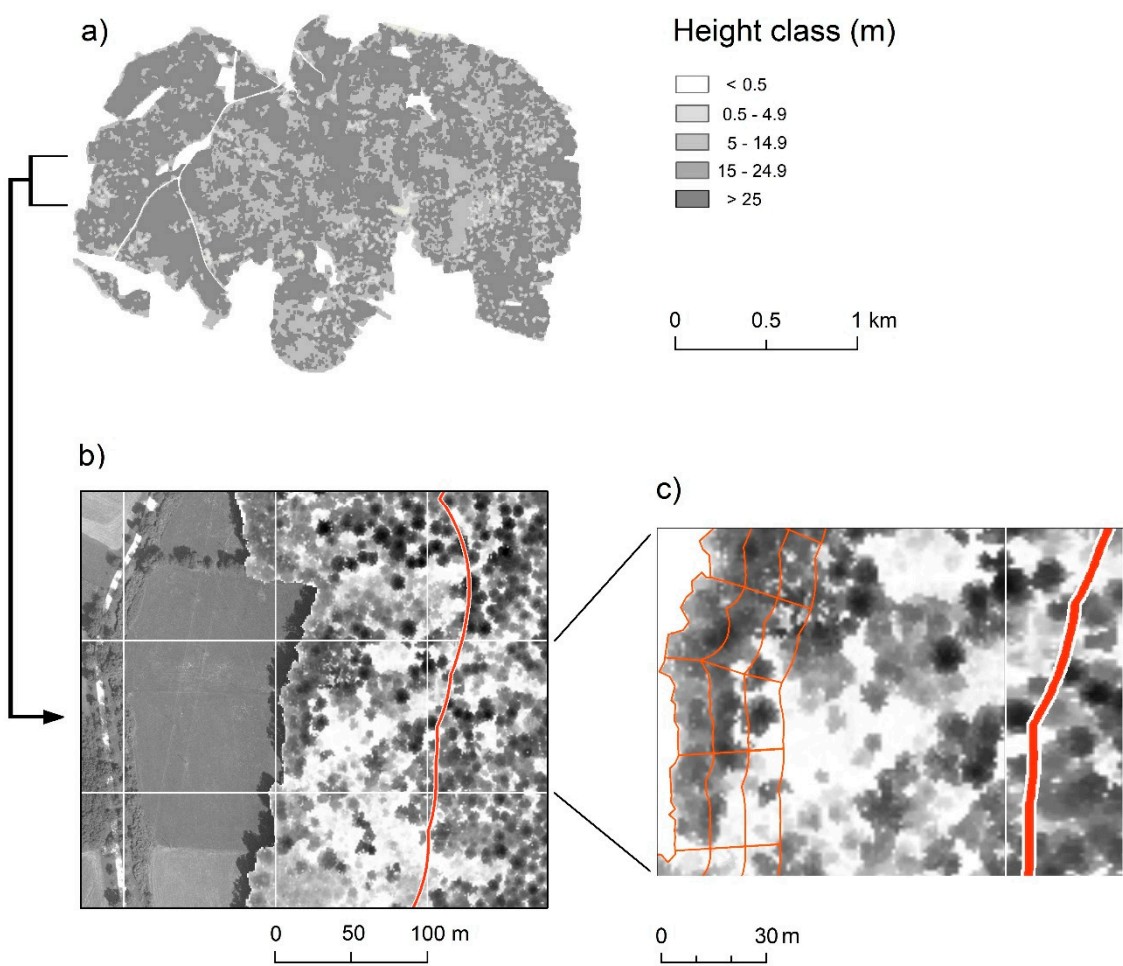

**Figure 3.** A schematic diagram of work methods. (**a**) Drawing a map of canopy height classes based on LiDAR data in the area of Rožnik forest patches; (**b**) An enlarged window depicting the examination of delimited forest edges in orthophoto images and determination of the sampling setup on the 100 × 100 m grid; (**c**) A box showing 30 m segments in 10, 20, and 30 m belts from the forest edge towards the interior forest area (**c**).

On the polygons of 30 × 10 m in the forest edge segments (Figure 3c), we used the canopy height model (CHK) in order to determine the heights of the tallest trees and pixel proportion of canopy cover at different height levels of trees as follows: above 0.5, 5, 15, and 25 m. From the proportion, we determined the vertical canopy height diversity (CHD, Equation (1)) in a similar way as in comparable studies of forest edges [31]:

$$CHD = -\Sigma p_i \times \ln (p_i) \tag{1}$$

where $p_i$ is the proportion of tree canopy at different height levels. We used the heights of the tallest trees and indices of canopy height diversity on different $30 \times 10$ m polygons as variables, which we subjected to a *k*-means cluster analysis [35]. The variables used for the assessment of the forest edge structure in the cluster analysis were polygon estimates composed of three 10 m distances from the forest edge to form a $30 \times 30$ m square. In forest inventories, we often define the width of the forest edge as the distance of an average mature tree height [30]. Our aim was similar, and therefore we decided to form groups of forest edges with different vertical shapes at a distance of a mature tree height.

We identified the optimal number of clusters by comparing the elbow method (total within-cluster variation is minimized) and the gap statistics method for determining optimal clusters [36] (R Core Team, 2014). The former method compares the total intracluster variation for different numbers of clusters with their expected values under null reference distribution of the data, with no obvious clustering [37] (Figure 4 and Table 1).

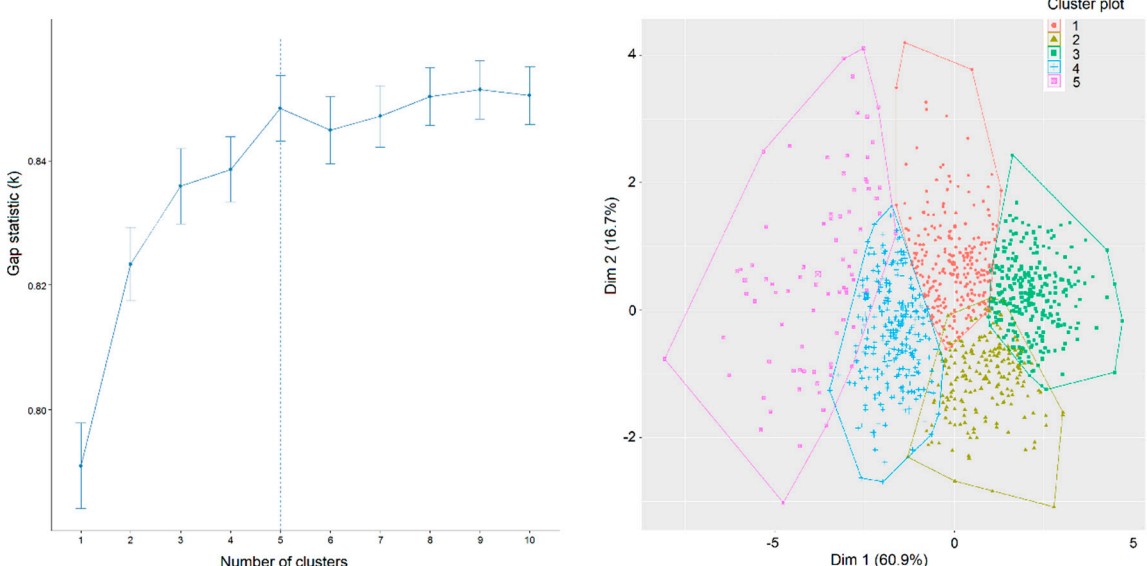

**Figure 4.** Determining optimal clusters on the forest edges in the study area of Nadgorica based on the data of the tallest trees and the proportion of canopy cover at different height levels of trees.

**Table 1.** Forest edge height classes in different clusters based on the data of the tallest trees in the study area of Golovec and Rožnik.

| Golovec | | | | | | Rožnik | | | | | |
|---|---|---|---|---|---|---|---|---|---|---|---|
| | Tree Height (m) | | CV (%) | | | | Tree Height (m) | | CV (%) | | |
| Cluster | Avg.$_{10}$ | SD$_{10}$ | 10 m | 20 m | 30 m | Cluster | Avg.$_{10}$ | SD$_{10}$ | 10 m | 20 m | 30 m |
| C1 | 31.8 | 3.5 | 10.8 | 9.1 | 11.3 | C3 | 30.7 | 3.3 | 10.8 | 9.5 | 9.9 |
| C4 | 28.7 | 2.9 | 10.2 | 8.7 | 11.4 | C4 | 28.7 | 2.7 | 9.3 | 11.5 | 16.9 |
| C3 | 25.8 | 2.9 | 11.3 | 10.5 | 12.1 | | | | | | |
| C5 | 23.6 | 2.4 | 10.4 | 11.0 | 13.3 | C5 | 23.6 | 4.0 | 16.8 | 11.7 | 12.9 |
| C6 | 17.8 | 3.9 | 21.7 | 21.1 | 12.4 | C1 | 21.4 | 4.8 | 22.3 | 19.2 | 26.5 |
| C2 | 17.7 | 3.9 | 22.2 | 24.0 | 29.1 | C2 | 6.4 | 2.9 | 45.0 | 116.6 | 9.3 |

Avg, arithmetic mean of the tallest trees on the forest edge polygons 10 m from the forest edge; SD, standard deviation; and CV, coefficient of variation in 10-m polygons 10, 20 and 30 m from the forest edge.

## 2.4. Fieldwork

We tested the characteristics of different clusters in the field. At the intersection of the map of forest edges with the $100 \times 100$ m systematic sampling grid (Figure 3b), we selected 30 m polygons, finding the tallest tree on each of them. Then, we determined its species, measured its height and its diameter at breast height (DBH), as well as estimated its canopy length (more than half of the total tree length, between one quarter and half of the total tree length, less than one quarter of the total tree

length). We estimated the heights of the trees measured in the field based on the canopy height model (CHM) data. Similar to a comparable research of the Slovenian forests [38], we estimated that LiDAR heights from low-density LiDAR are underestimated. On average, we estimated 1.3 m lesser heights for coniferous trees and 2.0 m for deciduous trees. We conducted height measurement in the field in the summer of 2019, five years after the LiDAR images were recorded. In the area of the systematically selected polygons, we estimated the characteristics of the forest edges as a transition zone between forest and other land uses. These estimates are based on the assessment of the forest edge as a shelter belt forming a barrier of trees and shrubs to provide protection from wind and storms. We adopted the assessment already tested by the Swiss National Forest Inventory [30] as follows:

1.  forest edge without shelter belt, without shrub belt;
2.  without shelter belt, with shrub belt;
3.  steep shelter belt, without shrub belt;
4.  protruding shelter belt (the branches of the edge trees extend into open land), without shrub belt;
5.  shrub belt mostly under the shelter belt;
6.  shrub belt clearly in front of the shelter belt;
7.  with shrub belt and a steeped, loose shelter belt.

We estimated their *h/d* (height/diameter) dimension ratio, based on the measured heights and diameters of trees. The size of the crown, crown length, and *h/d* ratio of a tree are considered important with respect to its stability during storms and snow [39,40]. The recommended *h/d* dimension ratio of trees is lower than 80, whereas in snow avalanche and rock fall protection forests, the values of coniferous trees should be lower than 65. In the growing of forests at gap edges, formation of long tree crowns should be supported.

## 3. Results

Peri-urban forests of Ljubljana differ from each other according to the degree of fragmentation represented by the share of the interior forest area and its depth. In two of the study areas, (Golovec and Nadgorica), the share of the forest interior that is more than 100 m away from the edges is greater than half (Table 2 and Figure 5d, Figure 6d, and Figure 7d). On the contrary, forest patches of Barje are characterized by their narrow linear shapes with the share of the interior area covering less than one fifth of the entire surface. Narrow belts of marshy meadows in the N–S direction break the only extensive forest patch in the area. The Barje forest patches developed from former hedgerows and waterside vegetation at drainage ditches on marshy lands.

**Table 2.** Surface of the peri-urban forest patches, the forest outer edge length, and the depth of the interior forest area.

| Forest Patches | Area (ha) | Outer Edge Length (km) | Interior Forest Area (%) | |
| --- | --- | --- | --- | --- |
| | | | >100 m | >300 m |
| Golovec | 672.53 | 34.79 | 63.3 | 23.4 |
| Rožnik | 345.68 | 17.06 | 46.7 | 8.1 |
| Nadgorica | 412.84 | 34.02 | 51.7 | 20.4 |
| Barje * | 453.14 | 72.62 | 17.9 | 0.7 |

* In the Barje area, only 50% of the lands covered with trees and woods are classified as forests. Alder coppices and poplar plantations belong to agricultural lands.

We conducted a thorough analysis of the edges of the persistent forest patches and estimated them with regard to the land use they border on. In the study area of the Golovec forest patch, more than half of the forest edge segments (55.8%) border on residential buildings, whereas in the areas of Rožnik and Nadgorica, the shares are smaller (19.1% and 10.2%, respectively).

In the study area of the Golovec forest patch, forest edge types notably differ regarding the heights of the tallest trees in the first 10 m forest edge belt (Figure 5a). The C3, C4, and C1 clusters belong to the

development stage of old trees. Among them, the C1 cluster stands out, with the average heights of the dominant trees in the first belt reaching 31.8 m (Table 1). In this patch, we decided to keep six clusters because of one of the differences we estimated in Clusters 2 and 6. In the first forest edge belt, both of these clusters have similar average heights (17.7 and 17.8 m). In the second and third 10 m forest edge belts, the heights of the tallest trees in the C2 cluster remain similar (18.0 and 18.3 m), whereas in the C6 cluster, the heights gradually increase to 22.3 and 25.8 m. These changes are represented in Figure 5b, which depicts the clusters on the opposite sides of the scale and also shows differences in the assessment of the proportion of tree canopy at different height levels (CHD). In the C6 cluster, the CHD values increase from 1.0 in the first 10 m belt to 1.3 in the second belt, and 1.4 in the last belt, whereas in the C2 cluster, the differences between the three 10 m forest edge belts regarding CHD values are small (1.0, 0.8, and 0.9). The C1 and C4 clusters represent the sharpest forest edges. In the C1 cluster, the CHD values are the same in all three belts (1.6) and the average heights of the dominant trees are also similar (31.8, 33.8, and 33.3 m). The C4 cluster differs from it not only regarding lower tree heights in all three belts (28.7, 29.4 and 27.8 m), but also lesser diversity of the proportion of tree canopy at different height levels CHD from 1.5 to 1.2, and 1.0).

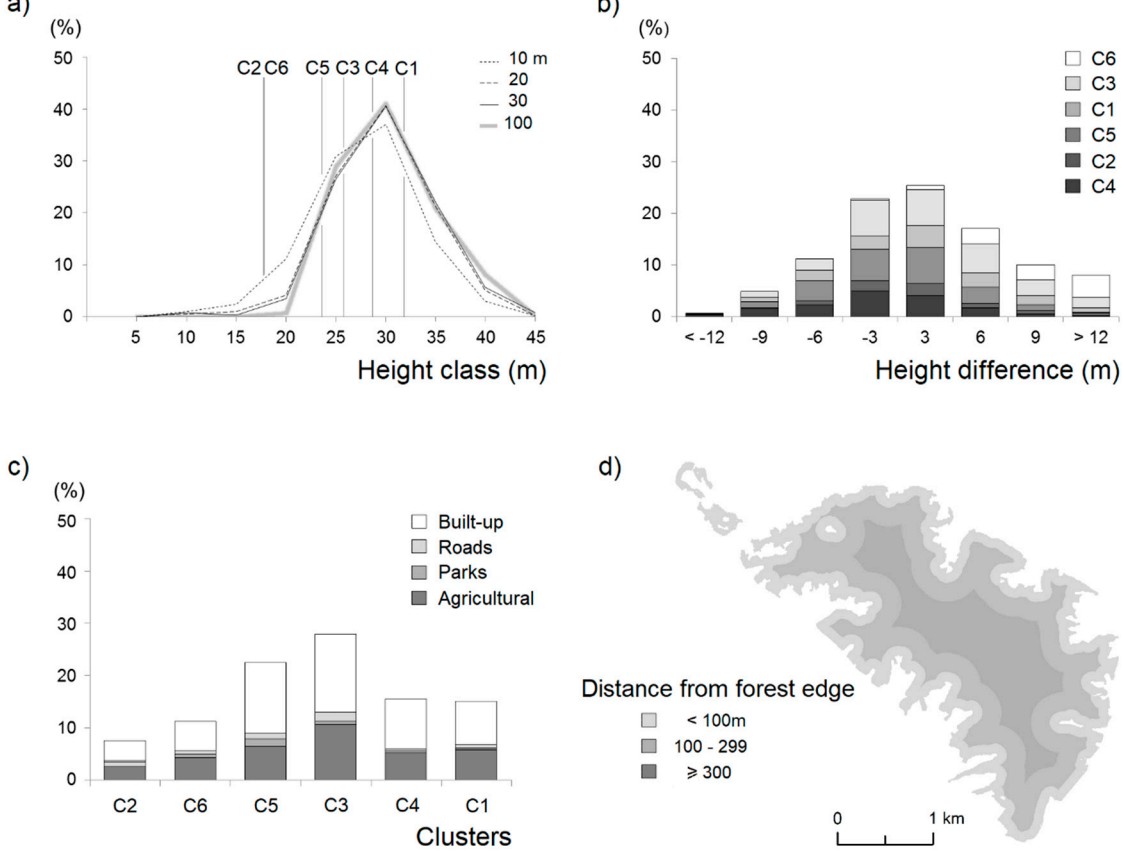

**Figure 5.** The proportion of canopy cover at different height levels of trees in 10 m polygons 10, 20, and 30 m from the forest edge and within a distance of 100 m in the direction of the interior forest area (**a**). Depicted are the average heights of the dominant trees in clusters at the forest edge (**a**); height differences between the tallest trees in the first and second forest edge belt (**b**); and the lands bordering on the forest edge (**c**); in the area of the Golovec forest patches (**d**).

In the study area of the Golovec forest patches, more than half of the forest edge segments (55.8%) border on residential buildings. However, the vertical structure of forest edges in the neighbourhood of settlements is the same as at the edges of agricultural lands (Figure 5c).

In the study area of Rožnik, 19.1% of forest edges border on built-up areas, including the highest and steepest edges of the forest patch. In the C3 cluster (Figure 6a, and Table 1), 45.7% of forest edges

border on settlements and one third of them on roads (Figure 6c), with the dominant trees in the three forest edge belts reaching the average heights of 30.7, 33.2, and 33.5 m. Graded forest edge was determined in the C2 cluster, its average tree heights gradually increasing, reaching 6.4, 11.6, and 29.4 m. It was estimated only on six forest edge polygons (Figure 6c). Additionally, graded structure is implied in the C5 cluster, in which the average heights of the dominant trees also increase, reaching 23.6, 27.1, and 29.8 m. Rožnik differs from the other patches, as the trees on its forest edge are taller than the trees measured within a 100 m distance in the direction of the interior forest area (Figure 6a).

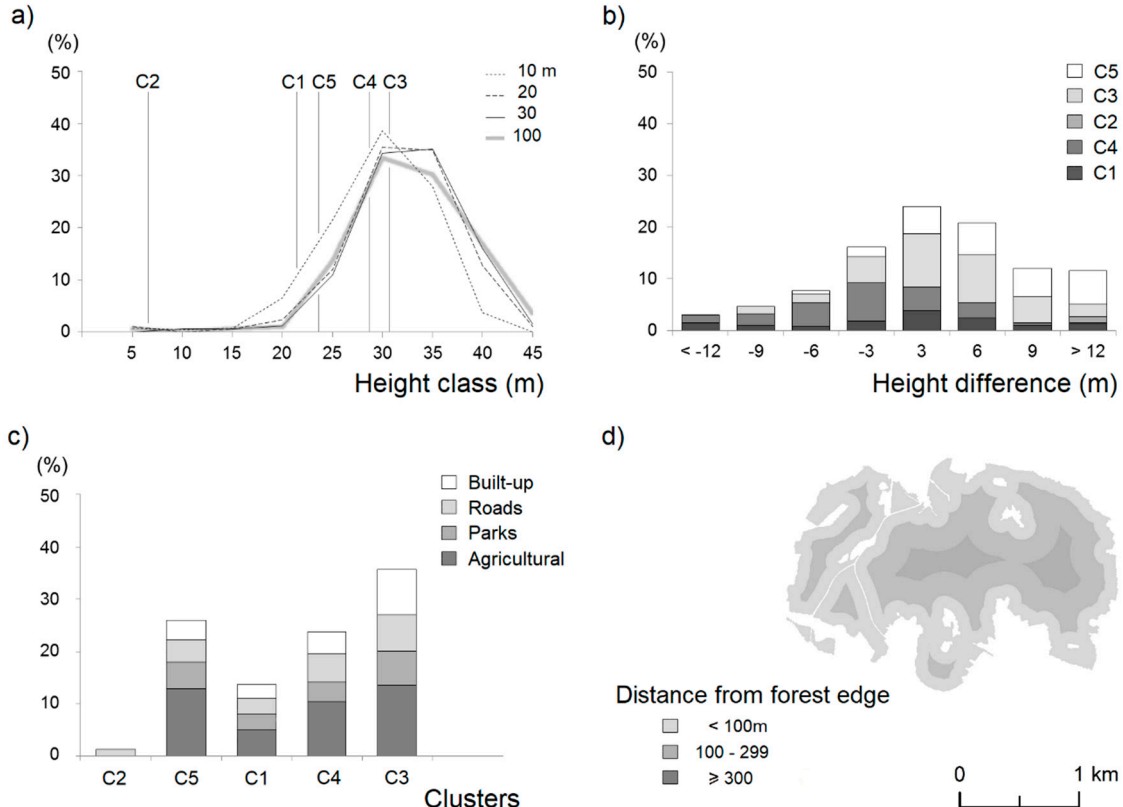

**Figure 6.** The proportion of canopy cover at different height levels of trees (**a**); height differences between the tallest trees in the first and second forest edge belts (**b**); and the lands bordering on the forest edge (**c**); in the area of the Rožnik forest patch (**d**).

In the study area of Nadgorica, the trees within the core area of forest patches are taller than the ones on the forest edges, which are at a younger development stage, as they mostly grew after the construction of a high-voltage power line and the formation of agricultural lands in its vicinity. Forest edge at the power line represents 28.0% of the entire edge. After the construction of the power line, some of the agricultural lands were overgrown by forest, while the others were cleared so that the trees at their edges would not endanger the power line network. We used orthophoto images from 1975 in order to determine the forest edges formed by the overgrowing of the abandoned agricultural lands in this area. Following the clearings in the area of the power line network, the most commonly formed edges were that of the C4 cluster (43.5%), which were also prevalent on the abandoned agricultural lands (41.5%).

Built-up areas border on 10.2% of forest edges, most frequently on the highest and steepest forest edges of the C2 and C3 clusters (Figure 7b,c).

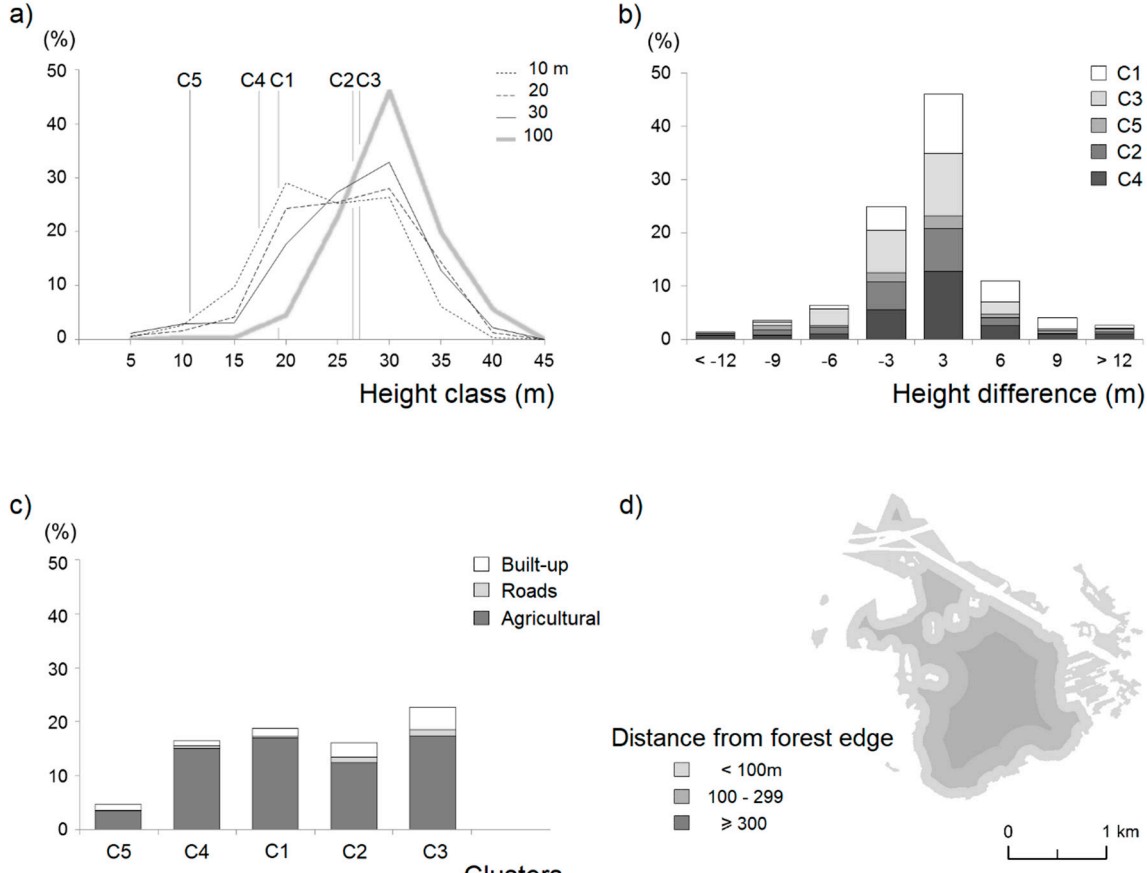

**Figure 7.** The proportion of canopy cover at different height levels of trees (**a**); height differences between the tallest trees in the first and second forest edge belts (**b**); and lands bordering on the forest edge (**c**); in the area of the Nadgorica forest patch (**d**).

When comparing orthophoto images from 1975 and 2018, we determined clearings for the benefit of residential houses only on 60 forest edge polygons with a combined length of 1800 m. Nowadays, the C2 and C3 clusters prevail on these edges, their common share exceeding two thirds (68.3%) of the surface. After the clearings at the edges of agricultural lands, only 32% of such forest edges were formed. We also determined similar characteristics of newly formed forest edges bordering on built-up areas in the area of Rožnik. Today, 62 forest edge polygons, cleared for the construction of buildings, have the highest, steep forest edges. The C3 and C4 clusters in Figure 6a represent 59.7% of forest edges bordering on built-up areas.

In the area of Golovec, we determined the highest share of forest edge clearings for the purpose of building (75%), with the combined length of 6.42 km. After the construction of buildings, 54% of the newly formed forest edges developed a high and steep structure (Figure 5, C1, C3 and C4 clusters). However, the characteristics of the dominant trees on forest edges cannot be determined based on the clusters they belong to alone. When conducting a field survey of forest edges, we found trees with unfavorable *h/d* dimension ratios in all the clusters.

In the area of Rožnik, the sample contained 7.3% of trees with an unfavorable dimension ratio, more among the coniferous than among the deciduous trees (Table 3). In the area of Golovec, the share of such trees was higher (16.2%), while trees with a crown shorter than half of the total tree length were also more numerous.

**Table 3.** The shares of the dominant tree species in a sample at the forest edges of Rožnik and Golovec, in the year 2019. The share of the trees with a crown shorter than half of the total tree length (CL) and the share of the trees with an unfavorable *h/d* dimension ratio for coniferous trees (>65) and deciduous trees (>80).

| Tree Species | Rožnik (%) | (*N* = 109) CL * (%) | *h/d* * (%) | Golovec (%) | (*N* = 191) CL * (%) | *h/d* * (%) |
|---|---|---|---|---|---|---|
| Conifers | 19.3 | 19.0 | 14.3 | 27.7 | 52.8 | 37.7 |
| *Picea abies* | 16.5 | 16.7 | 16.7 | 16.2 | 29.0 | 45.2 |
| *Pinus sylvestris* | 1.8 | | | 11.5 | 86.4 | 27.3 |
| Broad-leaves | 80.7 | 23.9 | 5.7 | 72.3 | 37.7 | 8.0 |
| *Fagus sylvatica* | 1.8 | | | 13.6 | 7.7 | 0.0 |
| *Quercus petraea* | 36.7 | 27.5 | 0.0 | 12.0 | 30.4 | 4.3 |
| *Acer pseudoplatanus* | 11.9 | 23.1 | 15.4 | 2.1 | | |
| *Fraxinus excelsior* | 6.4 | | | 0.5 | | |
| *Tilia sp.* | 6.4 | | | 2.6 | | |
| *Castanea sativa* | 3.7 | | | 13.1 | 25.0 | 8.0 |
| *Carpinus betulus* | 2.8 | | | 4.2 | | |
| *Robinia pseudoacacia* | 2.8 | | | 11.0 | 57.1 | 19.0 |
| *Alnus glutinosa* | 0.9 | | | 4.7 | | |

\* The share of the tree species was calculated if the sample contained at least 10 trees.

In addition to spruce and Scots pine, unfavorable *h/d* dimension ratios are common for sycamore maple, black locust, chestnut, and sessile oak. For trees with an unfavorable dimension ratio, we were unable to show its dependency on the length of their crowns, although among the coniferous trees at the forest edges of Golovec, we estimated 52.8% of coniferous and 37.7% of deciduous trees with crowns shorter than half of the total tree length (Table 3).

In the area of Golovec, the average *h/d* values of trees with short canopies were higher than the values of trees with canopies longer than half of the tree length ($P < 0.01$, t-test). In the area of Rožnik, the average *h/d* values were lower (Figure 8a), with an interesting difference in the average *h/d* ratio ($P < 0.01$) between the trees at the forest edge bordering on built-up areas and those in the neighbourhood of agricultural lands, parks, and other green areas (Figure 8b). On the basis of the comparison of the DBH of these trees, we estimate that in the area of forest edges bordering on built-up areas, trees at a younger development stage with an unfavorable *h/d* dimension ratio are prevalent (Figure 8c).

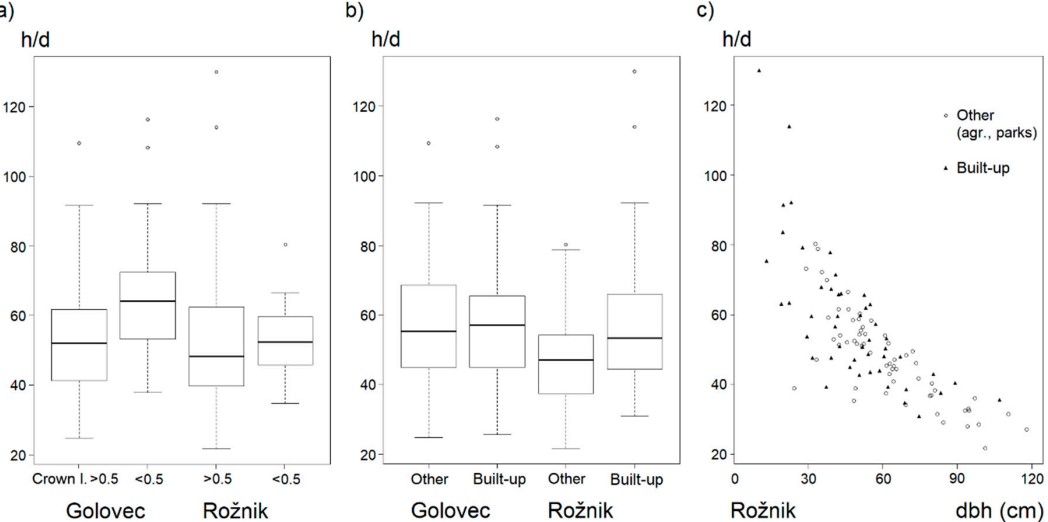

**Figure 8.** Box and whisker plots for the estimated *h/d* dimension ratio in the area of Golovec (*n* = 191) and Rožnik (*n* = 109) forest patches. (**a**) We estimated the dimension ratio based on the canopy length of the dominant tree in the segment; and (**b**) land use bordering on the forest edge (built-up, agricultural, parks, urban green areas, and other). (**c**) A diagram of the *h/d* dimension ratio regarding DBH for trees at the forest edges of Rožnik.

Forest edges with short-crowned trees are mostly filled by shrub belt. In the conducted field survey, we determined no presence of a developed shrub belt only in 8.4% of the estimated forest edge segments with short-crowned trees. The same share (8.3%) of such forest edges was determined in the area of Rožnik, where trees were estimated to have favorable dimension ratios (Figure 8). Forest edges have a developed shelter belt with shrub belt on 60.4% of the surface, while a further 23.6% is covered by forest edges without shrub belt, but with tree crowns forming a protruding shelter belt. In the area of Golovec, the share of the protruding shelter belt is smaller (6.8%), but 72.8% of the edges were estimated to have a developed shelter belt with shrub belt.

## 4. Discussion

For the management of peri-urban forests, we need detailed information on the structure of forest edges, which we can acquire from information systems of forest management planning or forest inventories. In their national inventories, many European countries have not yet gathered data on the structure of forest edges [17]. They are rarely evaluated separately on sampling plots or transects, which can form a basis for the assessment of their state on national or regional level [19]. When conducting reliable delimitation of land uses and their changes in monitoring systems, we often focus on the changes of the surface [34,41]. More rarely, we analyze transitions or ecotone areas between ecosystems typical of urban and sub-urban landscapes, and even then in connection with selected species [42]. The data we gather in the processes of spatial planning according to different sectors (nature preservation, forestry, agriculture, and urbanization) should be assessed based on the characteristics of human activities and processes in urban areas, be it due to infrastructure [16] or planning of new residential buildings at forest edges [43].

The tools and methods presented in the research allow the implementation of concrete cultivation practices within a realistic time frame, as we can ensure, based on a range of remote sensing data, that appropriate forestry measures are taken in the event of a disturbance.

The city of Ljubljana is surrounded by forests, except in the south where it borders on marshes. Consequently, every expansion of the city is bound to interfere with forest patches, changing them and forming new forest edges. In the preliminary research [22], we analyzed the influence of urbanization on forest connectivity. Now, we took a step further, focusing on the forest edge as the part of the forest first perceived and experienced by humans. On the basis of the impression of the forest edge, one evaluates the aesthetics of the entire forest patch. Furthermore, the edge of the forest is often its most vulnerable part due to human intervention and abiotic factors. Cultivation and preservation measures which are, in fact, initiated at the forest edge, can ensure the stability of the forest ecosystem, as well as the aesthetics and well-being.

Since clearcutting forest management is prohibited in Slovenia [44], extensive surface changes on the edges, as well as in the forest interior, are rare. While stationary forest edges have already been determined for the peri-urban forests of Ljubljana in preliminary studies, direct human influence on the shape of forest edges could only be inferred from a small share of changed forest edges, usually bordering on newly built-up areas. In the close-to-nature forest management, there is a tendency to imitate natural processes, in which small surface changes in forest structure play the most decisive role. Among the suggested changes, for instance, are formations of the smallest gap surface in an imitation of natural processes of beech rejuvenation, and 500 m$^2$ for a group rejuvenation. Experience shows that it is possible to ensure sustainable and quality beech rejuvenation by opening rejuvenation surfaces measuring from 0.1 to 0.2 ha, or from at least 0.25 ha in the case of oak at the first stages of rejuvenation, to 0.5 ha in the development phase of a pole timber [45]. Due to strict law-imposed limitations and the described forest management concept, significant changes of forest edge structure are caused only by natural disasters or biotic factors (insects, fungi). This results in a high share of steep forest edges in the central area of peri-urban forests (Golovec, Rožnik) (Figures 5 and 6). Since interventions in forest edges are rare, their vertical structure reaches that of the interior forest areas, as close as 10 m away

from the edge (Figures 5 and 6a). Similar to in the countries practicing intense forest management, graded forest edges are scarce [11].

Following the research on the development dynamics of forest edges, silviculturists suggested that the principle element of the borders remains a structure that permits the highest degree of irregularity while favoring rare and aesthetically interesting tree species. They warn that severe intervention at forest edge often has the effect of diminishing diversity due to the dominance of some pioneer shrubs [46]. Forest edges with a developed shrub belt were also prevalent in the areas of Golovec and Rožnik. This can be beneficial for the preservation of bird habitats [31], but it can also become an unsuitable obstacle in the area of urban forests. Graded forest edges are not always recommended, especially not in recreational areas or parks, where people prefer visually open forest edges giving them the sense of security [11]. In the urban forest of Rožnik, for instance, the trees at the forest edge are higher than those in the forest interior (Figure 6a). Despite the prevailing share of steep forest edges, we estimated tall trees with a favorable *h/d* dimension ratio. Another important observation we made was that at the forest edges near the buildings constructed in the last 40 years, trees at a younger development stage with an unfavorable dimension ratio are prevalent (Figure 8c).

Experience from other areas in Slovenia shows that urban construction often disturbs and fragments the forests. However, the replacement forest areas and the formation of forest edges after construction work has been completed have often been neglected [47]. As a consequence, the danger of unstable forest edges often remains, which also endangers the stability and protective function of such forests.

At younger development stages of such forest edges, a selective thinning approach is recommended, as it enables the possibility to accelerate or promote a range of tree species [48].

This poses a question on the structural changes influenced by cultivation measures and on how far into the interior forest area they reach. By comparing structural differences of forest edges at 10 m distances from the forest edge towards the interior forest area, we confirmed the findings of the researchers estimating that the structural changes in the forest edge are quite short, seldom exceeding 20 m [48]. In the area of Ljubljana, this applies to both groups of forest patches within the motorway ring road, namely Rožnik and Golovec. In both areas, the belts at 20 and 30 m from the forest edge had a similar height structure (Figures 5a and 6a). Only the first 10 m belt stood out regarding a higher share of lower tree heights, although graded forest edges were rarely estimated in the cluster analysis. The patches in Nadgorica (Figure 7) are an example of how the building of infrastructure (power line) causes the formation of new forest edges, which develop after large-scale human interventions. Several directions and guidelines are available for the management of such forest edges with emphasis on favourable dimension ratios (Figure 8), much like for roadside and railway corridors [11,16,48].

Alongside these corridors, we pass over to the estimation of forest edges on the landscape level. In the last ten years, the urbanization process in Slovenia has been most strongly influenced by the construction of the motorway network and the development of trade and industrial centers [49].

As shown in a study of 202 European cities [50], urban residential areas seem to grow independently of population changes. The role of urban nature and urban green infrastructure has become important to the members of today's urban society [51], since they benefit from urban and suburban green spaces providing them with physical and mental health and other "ecosystem services". Proper management of forest edges can be considered an early warning system ensuring the stability of forest patches in close-to-nature management.

Since LiDAR data are available to the majority of European countries [52], this data should be used not only to conduct research work, but also to develop methods and processes, which can be adopted by forestry experts in their operational work, especially in the field of urban forestry. The Canadian Forest Service application guide for generating an enhanced forest inventory [53] emphasizes that the ALS data and the application of this data to forestry have been the subject of active research for the last decades, while the operationalization of the technology in forest sector is a more recent phenomenon. To that end, the methodology and findings in the presented article are intended for the

operationalization of the ALS technology in urban forestry. We believe that the enhancement comes from the detailed information on forest edge structure and the increased spatial resolution derived from the ALS data in the area of peri-urban forests. In related studies of forest edges, the intention of the researches was primarily to evaluate the distance of influence of edge effects, measured on-site or using contiguous sampling forest bands for determining changes in vegetation composition of forests [30,54]. In addition, wall-to-wall mapping enables the grid-cell predictions to mean or other descriptive statistics of attributes of interest. The mapping of basic forest structural attributes requires a minimum of two sources of data, i.e., wall-to-wall ALS data for the forest management area of interest and ground plot measurements [53].

We presented a simple methodology which provides a starting point for monitoring of peri-urban forest surfaces and forest edges based on the processed LiDAR data freely available [28]. In the next monitoring cycles, we should be able to rely on the technology enabling the construction of a digital model of tree crowns based on image matching of aerial images. Similar to other European countries, Slovenia has an established system of cyclic aerial survey. The overlapping of stereoscopic aerial images already exceeds 80% [55], which enables the use of image matching technology [56]. On the basis of the data provided by the cyclic aerial survey of Slovenia, we determined it would not be necessary to replace the technology of laser scanning, as it is capable of exceptional resolution of ~450 points/m$^2$ and also it could possibly be affordable for large scale surveys in the coming years [57]. Using cutting edge ALS techniques, it is possible to acquire tree height measurements more reliably than using conventional field measurements. However, it is also important to develop methods enabling the use of ALS and other remote sensing technologies outside of study areas and operationalization of these technologies in forestry.

## 5. Conclusions

In forest inventories, ten-year periods are often chosen as time intervals for the assessment of forest changes and the efficiency of forest management. We made assumptions on the development of forest edges indirectly, based on their changes over the last decades. In the presented article, types of forest edges were determined based on cluster analysis, since the structure of forest edges has not yet been estimated by forest inventories, nor has forest management been given directives for cultivation measures on forest edges [58].

Within the city of Ljubljana, citizens strongly oppose interventions in forest edges similar to that across Europe, where the highest importance of the recreational value forest was attached to the size of trees within stands in older age classes [59,60]. While all forests perform many ecosystem services naturally, often urban and sub-urban forests also exhibit an emphasized recreation function. Consequently, it is important to consider related safety elements, namely the stand structure, especially at the forest edge where it enables a solid stability. In relation to this, we propose to establish monitoring systems of urban and sub-urban forests, especially within the depth of at least one stand height next to settlements, roads, and surfaces with an emphasized recreation function. Observations of the current state within the city of Ljubljana emphasize prevailing steep forest edges with tall trees. In the area of forest edges bordering on built-up areas, we estimated trees at a younger development stage, with an unfavorable dimension ratio.

Where the forest edge is open and there is enough space available, it is necessary to plant small tree species in order to build a stable forest edge foundation. With the LiDAR data, these parts can be reliably determined. Since forest edges have a developed shelter belt with shrub belt on more than 60% of the surface, such edges can become an unsuitable obstacle preventing the rejuvenation of trees by the dominance of some pioneer shrubs.

The recreational role of suburban forests is increasing, as is their importance for well-being, but the health issues (e.g., allergenicity) must not be overlooked. The northern part of the selected forest patch is characterized by a strong presence of the black alder, a highly allergic tree species. This species

was also estimated in the forest edge sample within the city of Ljubljana. This should be taken into consideration when maintaining the existing or planning new recreation paths.

The results of our study can be used to form a new, more objective estimation of climate and protection forest functions, taking into consideration both stationary forest edges, as well as those which could endanger human property or even life, in severe weather conditions (strong winds).

**Author Contributions:** Conceptualization, D.H. and J.P.; methodology, D.H.; software, D.H. and A.K.; validation, D.H. and J.P.; formal analysis, D.H.; investigation, D.H. and J.P.; resources, D.H. and A.K.; data curation, D.H.; writing—original draft preparation, D.H. and J.P.; writing—review and editing, D.H. and J.P.; supervision, J.P. and D.H. All authors have read and agreed to the published version of the manuscript.

**Funding:** This research received no external funding.

**Acknowledgments:** The authors are grateful to the anonymous reviewers for their helpful suggestions.

**Conflicts of Interest:** The authors declare no conflict of interest.

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
