# Peer review of "Evaluation of Forest Edge Structure and Stability in Peri-Urban Forests"

_forests, doi:10.3390/f11030338_

Round 1

Reviewer 1 Report

Dear authors,

I believe your research on the urban/peri-urban forests in the vicinity of Ljubljana is quite interesting. The growing importance of urban forestry makes studies such as this increasingly important.

I have made a number of recommendations and comments, embedded within the attached PDF. Below, I outline several general concerns, with the hope that these will help you improve your manuscript, and convey your research as clearly and persuasively as possible.

  1. From what I can understand from your 'Materials and Methods' section, your research design is sensible. However, the specifics of you how delineated forest edges and interiors are somewhat unclear to me (as noted within the PDF). I believe a generalized figure would be very beneficial for the reader's comprehension, and to ensure that your practices are well-supported by, and contribute to previous literature, as appropriate.
  2. What is the connection between your four study patches and the overall forests around Ljubljana? Do these four patches boarder surrounding forests? Are they similar in composition to other peri-urban forests? This information would help tie your research into studies of different cities.
  3. I think you do a good job of describing the results of your analysis for the four study patches in the 'Results' section, and you bring in relevant land use/management policy details in the 'Discussion' section. However, I do not see an adequate linkage of these two. In my view, you need to connect your findings to the broader research and management history more closely. I think your discussion section would be much stronger if you devote more of it to this.
  4. I think it is very nice to see your three recommended measures at the end of the document, but they're not entirely based on the content of this document (e.g. you did not look at a time series of lidar data to reveal changes in crown structure). So I think these recommendations should be tied more closely to your specific findings. Additionally, where you do refer to your findings, I think there is insufficient discussion, i.e. the third point concerning black alder. You mention that this species has a high abundance in one of the study area patches, but how specifically would you recommend that managers take this into consideration in the future?
  5. What is it about your methods that makes them superior to previous approaches in ensuring adequate forestry measures are taken to minimize disturbances? I see in the abstract, and briefly in the discussion, that you claim your approach enables cultivation practices in a realistic time period -- but are you comparing this to previous approaches that were unable to do this? What makes your approach superior, particularly in the context of disturbance? These themes should be much stronger in your discussion.

Author Response

To make it easier to keep up with the changes, we marked the lines where the reviewers had comments as line xy and in brackets as line new xy giving the place where the new manuscript contained the new solution according to the comment, after accepting the changes with the 'track changes' command.

We have decided not to list the corrections in the Discussion and Conclusions chapters line by line, but to suggest that the reviewers look at them as a new whole, as they are newly written and thus more logical and transparent for a second review.

Review 1:

From what I can understand from your 'Materials and Methods' section, your research design is sensible. However, the specifics of you how delineated forest edges and interiors are somewhat unclear to me (as noted within the PDF). I believe a generalized figure would be very beneficial for the reader's comprehension, and to ensure that your practices are well-supported by, and contribute to previous literature, as appropriate.                           

Answer: We have prepared new Figure (Figure 3) according to the suggestion

What is the connection between your four study patches and the overall forests around Ljubljana? Do these four patches boarder surrounding forests? Are they similar in composition to other peri-urban forests? This information would help tie your research into studies of different cities.

Answer: We have seprated chapter Discussion and Conclusions into two chapter: Discussion, Conclusions. We have significantly changed chapted Discussion also including an answer to your remark. 

I think you do a good job of describing the results of your analysis for the four study patches in the 'Results' section, and you bring in relevant land use/management policy details in the 'Discussion' section. However, I do not see an adequate linkage of these two. In my view, you need to connect your findings to the broader research and management history more closely. I think your discussion section would be much stronger if you devote more of it to this.     

Answer: We have significantly changed chapter Discussion also including an answer to your remark. 

I think it is very nice to see your three recommended measures at the end of the document, but they're not entirely based on the content of this document (e.g. you did not look at a time series of lidar data to reveal changes in crown structure). So I think these recommendations should be tied more closely to your specific findings. Additionally, where you do refer to your findings, I think there is insufficient discussion, i.e. the third point concerning black alder. You mention that this species has a high abundance in one of the study area patches, but how specifically would you recommend that managers take this into consideration in the future?                         

Answer: We have significantly changed chapter Results and Discussion also including an answer to your remark. Numerous comments have been added based on the findings in the results section. A new Figure 8 is added which is relevant for the assessment of land use / management policy measures.

We added aditional text  (in new line 138):

The four described forest patches are composed of tree species that also build peri-urban forests in the wider study area (Fig. 1) and are characteristic of the prevalent forest site types .

What is it about your methods that makes them superior to previous approaches in ensuring adequate forestry measures are taken to minimize disturbances? I see in the abstract, and briefly in the discussion, that you claim your approach enables cultivation practices in a realistic time period -- but are you comparing this to previous approaches that were unable to do this? What makes your approach superior, particularly in the context of disturbance? These themes should be much stronger in your discussion.                                     

Answer: we added aditional explanation in Discussion (new line 548):

In Canadian Forest Service application guide for generating an enhanced forest inventory [53], it was emphasized that the ALS data and their application to forestry have been the subject of active research for the last decades, while the operationalization of the technology in forest sector is a more recent phenomenon. To that end, the methodology and findings in the presented article are intended for the operationalization of the ALS technology in urban forestry. We believe that the enhancement comes from the detailed information on forest edge structure and the increased spatial resolution derived from the ALS data in the area of peri-urban forests. In related studies of forest edges, the intention of the researches was primarily to evaluate the distance of influence of edge effects, measured on-site or using contiguous sampling forest bands for determining changes in vegetation composition of forests [30, 54]. On the other hand, wall-to-wall mapping enables the grid-cell predictions to mean or other descriptive statistics of attributes of interest. The mapping of basic forest structural attributes requires a minimum of two sources of data: wall-to-wall ALS data for the forest management area of interest and ground plot measurements [53].

Line 11 (new line 10): Country name.

Answer: in the presented research, we studied the forest edge structure of a peri-urban forests in 10 the outskirts of Ljubljana (Slovenia) consisting of a number of patches covering the collective surface of 1884 11 ha.

Line 19 (new line 18): Vertical canopy diversity

Answer: canopy hight diversity (CHD).

Line 54 (new line 53): extend                      

Answer: extent

Line 69 (new line 68): was

Answer: were

Line 79 (new line 78): accessible as forests

Answer: accessible than forests

Line 81 (new line 80): horizontal position

Answer: These forest edges can be classified as stationary, since edge displacement has not been observed and is therefore permanent for the time being.

Lines 88 and 90 (new line 87 and 89): grammar and scientific remark

Answer: We have analysed the condition of forest edges in the areas of the selected peri-urban forest patches and compare how different land uses and human interventions leading to forest edge clearings influence their structure. We have compared…. Structural characteristics of the forest edges were assesed using data from laser scanning (ALS), ciclic aerial surveys over 40-year period and field work on a systematic sampling grid.

Line 103 (new line 107): significant

Answer: important

Lines 116 – 118 (new lines 121 - 125: include scientific species names

Answer: … are beech Fagus sylvatica  (25%), sessile oak  Quercus petraea (23%), scots pine Pinus sylvestris (22%), chestnut  Castanea sativa (13%) and spruce Picea abies (11%). The Rožnik forest patch represents the majority of a landscape park dominated by spruce Picea abies (37%), scots pine Pinus sylvestris (16%), sessile oak Quercus petraea (14%) and chestnut Castanea sativa (11%)….

Line 119 (new line 125): define both patches

Answer: Both patches, Golovec and Rožnik indicate certain green corridor possibilities in the NW-SE direction.

Line 120 (new line 126): removing word 'early'

Answer: … landscape units [21], recognized as …

Line 151 (new line 160): avoid the term 'accurate'

Answer:  … to obtain an estimate of …

Line 154 (new line 163): Height classes and vegetation components by height classes above ground level defined as ≥ 0.5 m.

Answer: In national forest inventories [30] woody plants higher than 0.5 m are considerd as a part of the shrub belt. Dwarf shrubs and woody plants < 0.5 m are part of the herb border, similar to the estimation of vegetation structure in fragmented woodlands and their edges [31].

Line 165 (new line 174): describe potential positional accuracy problems

Answer: The older panchromatic black-and-white (PAN) aerial film photographs taken in scale of 1:17,500 were scanned using 21 µm resolution to achieve comparable resolution or ground sampling distance (GSD)  of 50 cm. The absolute orientation of the images was based on at least 10 to 15 ground control points. Since we acquired control points from topographic maps, we could only achieve a positional accuracy of 2-3 m for orthophoto images.

Line 177 (new line 193): 30m long (description)

Answer: We have prepared new Figure (Figure 3) showing the length of the individual segments.

Line 180 (new line 196): acquitted

Answer: acquired

Line 190 (new line 206): why certain distances were chosen (e.g. depth of interior area)

Answer: In several scientific papers a minimum patch size of 30–40 ha has been chosen in studies of habitat fragmentation for the protection of forest-interior species at the landscape scale.  A circular patch with such an area would have a radius of at least 300 m

Line 205 (new line 229): I would lead with this definition

Answer: We have restructured the paragraph in accordance with the reviewer's comment: In forest inventories we often define the width of the forest edge as the distance of an average mature tree height [36]. Our aim was similar therefore we decided to form groups of forest edges with different vertical shapes at a distance of a mature tree height

Line 210 (new line 238): Refer to Figure 3

Answer: … with no obvious clustering [37] (Fig. 4)

Line 249 (line 280): classes numbers?

Answer: we added class numbers

Line 255 (new line 288): Spell out the terms

Answer: h/d (height/diameter) dimension

Line 264 (new line 297): names

Answer: In two of the study areas (Golovec and Nadgorica), the share of the forest interior more than 100 m away from the edges is greater than half (Table 2, Figures 5d, 6d and 7d

Line 267 (new line 300): images of other study areas ?

Answer: we have prepared new figures (see fig. 3, 5d, 6d and 7)

Line 293 (new line 325): sentence explanation

Answer: In this patch, we decided to keep 6 clusters because of one of the differences we estimated in clusters 2 and 6.

Line 306: figure resolution

Answer: The images have been replaced with new ones in which we present the depths of the inner environment suggested by the reviewer at line 267.

Line 393 (new line 427): scots

Answer: Scots

Line 395: confusing phasing

Answer: A new Figure 8 image was made and a test confirming differences between trees with shorter and longer crowns (P <0.01).

Line 411 (new line 460): edges and inventories

Answer: In their national inventories, many European countries have not yet gathered data on the structure of forest edges [17]. They are rarely evaluated separately on sampling plots or transects, which can form a basis for the assessment of their state on national or regional level [19].

Line 411 (new line 460): specific species

Answer: selected species

Line 423 (new line 470): What are previous monitoring methods, and why are the insufficient? What makes your approach better for detecting or monitoring disturbances?

Answer: following part will be deleted:  some of which will be further discussed in this chapter. Whole chapter is rewritten

Line 431 (new line 566): 80% of what?

Answer: explantion is given:

The overlapping of stereoscopic aerial images already exceeds 80% [44], which enables the use of image matching technology.

Line 443: include a more comprehensive discussion of how your specific findings relate to these policies and land use histories

Answer: Discussion was changed. Numerous comments have been added based on the findings in the results section. A new Figure 8 is added which is relevant for the assessment of land use / management policy measures.

Line 463: avoid 'accurately'

Answer: these parts ov Conclusions have been changed.

Reviewer 2 Report

In this research, the authors characterize the structure of urban forest patches in Ljubljana, Slovenia. They use LiDAR-based analysis, supplemented with field inventory, to map forest patches based on their canopy height and analyzed them based on their spatial orientation in the semi-urban environment. Based on their results, they present guidance for future urban canopy development/conservation.

The study is rather straight-forward and is overall a pleasant read. However, there some issues that will need to be addressed before I can recommend it for publication:

Major Comments:

  1. I found myself losing the narrative at times, especially the methodology. I believe the paper would greatly benefit from a schematic diagram that demonstrates how all the pieces fit together (the LiDAR, clusters, field data, change analysis, etc.). I also think part of this is due to the writing style. There are way too many run-on sentences and commas are used too heavily. The authors should be more direct in their writing and limit the use of the passive voice.
  2. The authors repeatedly describe the mapping methodology as “accurate” (lines 151, 413, 463), and, while I do not disagree that the data is likely accurate, that is never demonstrated in the text. Sources of uncertainty or bias should be at least discussed, and it would help if the authors compared field-based height estimates to the LiDAR.
  3. The research is framed in the context of studying ecosystem services of urban forests (line 83), but it’s really a study on the extent and structure of forest patches in Ljubljana. That information can be used to assess ecosystem services but is not a service in its own right. The authors should be clearer about the objectives of the study so they are not overstating the results of their research.
  4. This might not be a major concern, but readers never get to see a map of the forest traits within the patches. Notably, a simple map of the forest height classes and clusters would be useful for understanding the results. This would also address my first concern about losing the “narrative”.

Line-by-Line

38: Can you define what peri-urban means?

46: Who is we?

81: “horizontal position” is a strange phrase. I’d suggest “spatial extent”

102: I don’t understand the use of the descriptor “wide”. In what context? I’d suggest just removing it and simply calling it the study area.

105: Please define a.s.l.

136: This is an example of a sentence can be reworded to not include a comma and be more direct.

238: What grid? Did I miss this some where?

258: Are avalanches occurring here?

420: This is an example of a run-on sentence that is hard to follow. Also, “chapter” is not the right word to use here – “section” would be more appropriate.

Figures:

Can you include a regional/country-scale map as a subplot in Figure 1?  

The font size should be increased on figures 3-6.  

Some captions require more description. For example, none of the maps (d in 4, 5, 6) are explained at all. The figures should be understandable outside the context of the paper.

The maps in 4,5,6 would benefit from a basemap or background imagery to give context.

Figures 5 and 6 are missing subplot labels.

Author Response

To make it easier to keep up with the changes, we marked the lines where the reviewers had comments as line xy and in brackets as line new xy giving the place where the new manuscript contained the new solution according to the comment after accepting the changes with the 'track changes' command.

We have decided not to list the corrections in the Discussion and Conclusions chapters line by line, but to suggest that the reviewers look at them as a new whole, as they are newly written and thus more logical and transparent for a second review.

Review 2:

I found myself losing the narrative at times, especially the methodology. I believe the paper would greatly benefit from a schematic diagram that demonstrates how all the pieces fit together (the LiDAR, clusters, field data, change analysis, etc.). I also think part of this is due to the writing style. There are way too many run-on sentences and commas are used too heavily. The authors should be more direct in their writing and limit the use of the passive voice.        

Answer:   We prepared new Figure 3 explaing the demand. We have provided language editing of the text

The authors repeatedly describe the mapping methodology as “accurate” (lines 151, 413, 464), and, while I do not disagree that the data is likely accurate, that is never demonstrated in the text. Sources of uncertainty or bias should be at least discussed, and it would help if the authors compared field-based height estimates to the LiDAR.                                      

Answer: line 151 (new line160): obtain an estimate            line 413 (new line 463):reliable , line 464 (new line 596): rewritten text

We also provided following text in new line 270:

We estimated the heights of the trees measured in the field based on the canopy height model (CHM) data. Same as in a comparable research of the Slovenian forests [38], we estimated that LiDAR heights from low-density LiDAR are underestimated. On average, we estimated 1.3 m lesser heights for coniferous trees and 2.0 m for deciduous trees. We conducted height measurement in the field in the summer of 2019, five years after LiDAR images were recorded.

The research is framed in the context of studying ecosystem services of urban forests (line 83), but it’s really a study on the extent and structure of forest patches in Ljubljana. That information can be used to assess ecosystem services but is not a service in its own right. The authors should be clearer about the objectives of the study so they are not overstating the results of their research. 

Answer: we added new text in new line 87: In this paper, we have focused on the structure of the forest edge which can significantly affect the sustainability of the protective and climatic functions of the forest.

This might not be a major concern, but readers never get to see a map of the forest traits within the patches. Notably, a simple map of the forest height classes and clusters would be useful for understanding the results. This would also address my first concern about losing the “narrative”.

Answer: We have prepared new figures (see Fig 3)

Line-by-Line

38: Can you define what peri-urban means?        

Answer: we have replaced with better known term Sub-urban. However both terms are well known and used in urban forestry texsts.

46 (new line 45): Who is we?                                                                     

Answer: we have replaced 'we' with term 'researchers'

81 (new line 80): “horizontal position” is a strange phrase. I’d suggest “spatial extent”

Answer: Paragraph has been rewritten also in accordance with Reviewer1: These forest edges can be classified as stationary, since edge displacement has not been observed and is therefore permanent for the time being.

102 (new line 106): I don’t understand the use of the descriptor “wide”. In what context? I’d suggest just removing it and simply calling it the study area.

Answer: we have removed word 'wide'                                                                                                

105 (new line 109): Please define a.s.l.                                    

Answer: a.s.l. (above sea level)

136 (new line 145): This is an example of a sentence can be reworded to not include a comma and be more direct.

Answer: We estimated the structure of stands and forest edge using LiDAR records which are freely available and cover the whole of Slovenia.

238: What grid? Did I miss this some where?

Answer: see in a new Figure 3 (line 210)

258 (new line 291): Are avalanches occurring here?                          

Answer: it is a general explanation of an important issue for protection forests role.

420 (new line 470): This is an example of a run-on sentence that is hard to follow. Also, “chapter” is not the right word to use here – “section” would be more appropriate.

Answer: The tools and methods presented in the research allow the implementation of concrete cultivation practices within a realistic time frame, as we can ensure, based on a range of remote sensing data, that appropriate forestry measures are taken in the event of a disturbance.

Figures:                                                                              

Can you include a regional/country-scale map as a subplot in Figure 1? 

Answer: We have prepared new Figures

The font size should be increased on figures 3-6. 

Answer: We have prepared new Figures

Some captions require more description. For example, none of the maps (d in 4, 5, 6) are explained at all. The figures should be understandable outside the context of the paper.

Answer: We Have prepared new Figures and aditional explanatory text: In two of the study areas (Golovec and Nadgorica), the share of the forest interior more than 100 m away from the edges is greater than half (Table 2, Figures 5d, 6d and 7d).

The maps in 4,5,6 would benefit from a basemap or background imagery to give context.

Answer: The images have been replaced with new ones in which we present the depths of the inner environment suggested by the reviewer1.

Figures 5 and 6 are missing subplot labels.

Answer: we have prepared new figure 5d, 6d and 7d with subplot labels.

Reviewer 3 Report

The manuscript contains signifiant research in urban forestry.

The information of paper is valuable for future management of urban and peri-urban forests. 

Thank you for good job!

Author Response

To make it easier to keep up with the changes, we marked the lines where the reviewers had comments as line xy and in brackets as line new xy giving the place where the new manuscript contained the new solution according to the comment after accepting the changes with the 'track changes' command.

We have decided not to list the corrections in the Discussion and Conclusions chapters line by line, but to suggest that the reviewers look at them as a new whole, as they are newly written and thus more logical and transparent for a second review.

Reviewer_3:

Thanks for appreciating our research and for finding it useful in urban forestry.

Round 2

Reviewer 1 Report

Thank you for carefully responding to the reviewer comments and suggestions. I feel that with the improvements you have made, your article is substantially improved. Congratulations on your interesting research.

Author Response

Reviewer1:

Thank you for carefully responding to the reviewer comments and suggestions. I feel that with the improvements you have made, your article is substantially improved. Congratulations on your interesting research.

Answer:

Thank you for supporting our work with usefull suggestions.

Reviewer 2 Report

I would like to thank the authors for addressing my concerns. The readability of the manuscript was improved and the added/altered figures are helpful. 

I have three minor suggestions regarding figures 5-7. First, there are no figure labels in figures 6-7, although they are referred to in the text and caption (such as a,b,c,d). Second, I still find myself squinting my eyes to read the text in the figures, especially in the legends. Finally, I don't see how the maps in these figures add to the manuscript. They are hard to make sense of without any context. I don't see anything scientifically inappropriate about them, however. 

Author Response

Reviewer2:

I have three minor suggestions regarding figures 5-7. First, there are no figure labels in figures 6-7, although they are referred to in the text and caption (such as a,b,c,d). Second, I still find myself squinting my eyes to read the text in the figures, especially in the legends.

Answer:

We have included all suggestions from reviewers and editor into a new version of our text

('manuscript3'). We have prepared new figures 5, 6, 7, with enlarged titles and legends. You will find these three figures included into 'manuscript3' and sent separately as a zip file.

Reviewer2:

Finally, I don't see how the maps in these figures add to the manuscript. They are hard to make sense of without any context. I don't see anything scientifically inappropriate about them, however.

Answer:

We do not agree with this statement. According to our view we conneted figures with revised manuscript in several places as follows (new lines in Manuscript3 after you will have accepted changes):

New line 198:

Our aim was to determine whether the canopy structure of forest edges differs from the canopy structure of forest stands in the interior area of forest patches.  We selected the distances of 100 and 300 m based on the suggested recommendations argued in the previous studies [35,22].

New line 291:

Peri-urban forests of Ljubljana differ from each other according to the degree of fragmentation represented by the share of the interior forest area and its depth. In two of the study areas (Golovec and Nadgorica), the share of the forest interior more than 100 m away from the edges is greater than half (Table 2, Figures 5d, 6d and 7d). 

New line 304 (in Table 29:

Interior forest area (%) 

> 100 m > 300 m from forest edge

New line 352:

Rožnik differs from the other patches, as the trees on its forest edge are taller than the trees measured within the 100-m distance in the direction of the interior forest area (Fig 6a).

New line 490:

Since interventions in forest edges are rare, their vertical structure reaches that of the interior forest areas as close as 10 m away from the edge (Figure 56a). Same as in the countries practicing intense forest management, graded forest edges are scarce [11].
